# Phytogenic Effects on Layer Production Performance and Cytoprotective Response in the Duodenum

**DOI:** 10.3390/ani13020294

**Published:** 2023-01-14

**Authors:** Evangelos C. Anagnostopoulos, Ioannis P. Brouklogiannis, Eirini Griela, Vasileios V. Paraskeuas, Konstantinos C. Mountzouris

**Affiliations:** Laboratory of Nutritional Physiology and Feeding, Department of Animal Science, School of Animal Biosciences, Agricultural University of Athens, Iera Odos 75, 11855 Athens, Greece

**Keywords:** phytogenic, nutrigenomic, laying hen, performance, gut function

## Abstract

**Simple Summary:**

Stressor challenges can lead to oxidative stress that can negatively impact poultry health and production. Phytogenics may exert beneficial effects on performance and product quality and enhance the endogenous antioxidant system. Therefore, the aim of this study was to evaluate the effects of dietary inclusion levels of a phytogenic premix on the production performance and critical detoxification and antioxidant molecular responses in the duodenum of laying hens. The overall laying rate and egg mass were significantly increased by phytogenic premix inclusion compared with the non-supplemented control. Phytogenic inclusion beneficially affected the expression of critical genes related to detoxification and antioxidant capacity. In conclusion, using an analytical nutrigenomics approach, this study provides new knowledge that further supports the noted phytogenic benefits for layer production performance. This knowledge could provide a new basis for diet formulation strategies using phytogenics in commercial conditions with multiple stressor challenges.

**Abstract:**

The aim of this study was to evaluate the effects of a phytogenic premix (PP) on the production performance and critical genes relevant to the detoxification (i.e., aryl hydrocarbon receptor pathway) and antioxidant (i.e., nuclear factor erythroid 2-related factor 2 pathway) response in the duodenum of laying hens. The PP was based on bioactive substances derived from ginger, lemon balm, oregano, and thyme (Anco FIT-Poultry). A total of 385 20 week old Hy-Line Brown layers were assigned to five dietary treatments with seven replicates of 11 hens each for a 12-week feeding trial. The experimental treatments included a corn–soybean meal basal diet with no PP (CON) or supplemented with PP at 500 (P500), 750 (P750), 1000 (P1000), and 1500 mg/kg diet (P1500). The overall (1–12 weeks) laying rate (*p* < 0.001) and egg mass (*p* = 0.008) were significantly increased in the P1000 group compared with the CON. At the duodenum, increasing dietary PP inclusion levels beneficially affected (*p* ≤ 0.05) the expression of the majority of the *AhR* and *Nrf2* pathway genes studied. In conclusion, according to the gene expression analysis, PP inclusion resulted in a reduced requirement for detoxification and an increased antioxidant capacity, with most of the effects seen at the PP inclusion range of 750 to 1000 mg/kg diet.

## 1. Introduction

The poultry industry makes a substantial contribution to food security and nutrition. Stressor challenges due to environmental (e.g., temperature, feed composition, and xenobiotics), microbiological, and management factors can lead to oxidative stress. The latter, if not appropriately controlled, can negatively impact poultry health and production [1]. 

Diet exposes birds to a wide variety of stress factors (e.g., xenobiotics) which may in turn affect their health and productivity. In order to minimize undesirable factors, diets may contain bioactive compounds conferring an additional cytoprotective effect by stimulating the innate detoxifying and antioxidant defense system. The latter involves the inducible gene expression of cytoprotective proteins with detoxifying, antioxidant, and anti-inflammatory functions [2]. Feed additives such as phytogenics (e.g., aromatic plants, herbs, spices, essential oils, and their bioactive components) have been shown to exert beneficial effects on performance and product quality and also enhance the endogenous antioxidant system [3,4].

More specifically, dietary phytogenics activate two signaling pathways referred to as *AhR* and *Nrf2*, namely, the aryl hydrocarbon receptor (*AhR*) and the nuclear factor-erythroid-derived 2-like 2 (*Nrf2*) [5]. In particular, the transcription factors *AhRs* are responsible for the regulation of target genes related to the detoxification and elimination of xenobiotic compounds such as dioxins, mycotoxins, phytochemicals, and bacterial pathogens [6,7]. On the other hand, the transcription factor *Nrf2* is responsible for the regulation of the antioxidant response and inflammation [8,9]. 

So far, limited studies have dealt with the *AhR* and *Nrf2* pathway modulation in laying hens at the intestinal level. For example, the upregulation of *Nrf2* in the jejunum of laying hens by dietary phytogenics was correlated with reduced oxidative stress [4]. Moreover, when star anise was supplemented in the diet of laying hens, certain *Nrf2* pathway (i.e., *SOD, CAT, GSH-Px*) genes were upregulated, resulting in an enhanced antioxidant capacity in the liver and serum [10]. It is also known that the efficacy of a phytogenic, among other factors, is related to the chemical properties of its constituents and the inclusion levels in the diet [11,12].

Therefore, the aim of this study was to evaluate the effects of dietary inclusion levels of a phytogenic premix (PP) on the production performance and critical detoxification (*AhR* pathway) and antioxidant (*Nrf2* pathway) molecular responses in the duodenum of laying hens.

## 2. Materials and Methods

### 2.1. Animals, Housing, and Experimental Treatments

The birds were received at 13 weeks of age. After a 7-week rearing period, 385 commercial Hy-Line Brown laying hens (20 week old), with uniform body weight and similar performance (without significant differences between the treatments), were allocated into 5 treatments with 7 replicates of 11 hens each for a 12-week experimental period (from 21 to 32 weeks of layer age). Dietary treatments included: control (C) basal diet without PP addition and basal diets supplemented with PP at 500 (P500), 750 (P750), 1000 (P1000), and 1500 (P1500) mg/kg. The PP used in this trial (Anco FIT-Poultry, Anco Animal Nutrition Competence, GmbH, Sankt Poelten, Austria) was a proprietary mixture of phytogenic substances marketed as a “gut agility activator” with an active ingredient concentration of 70 g/kg. The PP consisted of bioactive substances derived from ginger, lemon balm, oregano, and thyme. All diets were formulated to meet or exceed the recommendations provided in the Hy-line Brown Management Guide (2018). Feed, in mash form, and water were provided ad libitum during the experiment. Hens were kept in 3-floor battery cages (12 cages per floor) under controlled environmental conditions, and a gradual increasing light photoperiod was maintained up to 30 weeks of age, until 16L:8D. The air temperature was adjusted at 21–25 °C and humidity at 40–60% according to the manual of the Hy-Line Brown Management Guide.

### 2.2. Layer Growth Performance Responses

Eggs produced were collected and weighed daily. The number of the eggs and average egg weight were recorded. The laying rate was determined each week as the total number of eggs divided by 7 days. Feed intake was recorded on a weekly basis. Egg mass was calculated by multiplying average egg weight by laying rate. Feed conversion ratio was calculated as grams of feed intake per gram of egg mass. 

### 2.3. RNA Isolation and Quantitative Real-Time PCR

At the end of the 8th and the 12th week, 7 hens per treatment were randomly selected, anaesthetized (EC 1099/2009), and euthanized by severing the jugular vein, and samples from the duodenum were carefully excised aseptically, snap-frozen in liquid nitrogen, and subsequently stored at −80 °C for further analysis.

Τhe central section of the duodenum was exposed and the segments without digesta were washed completely in 30 mL cold phosphate-buffered saline (PBS)–ethylene diamine tetra-acetic acid (EDTA; 10 mmol/L) solution (pH = 7.2). Eventually, the total RNA from the duodenum was obtained according to the manufacturer’s protocol (Macherey-Nagel GmbH & Co. KG, Duren, Germany) with the NucleoZOL reagent. RNA quantity and quality were ascertained by spectrophotometry with the NanoDrop-1000 by Thermo Fisher Scientific, Waltham, United Kingdom. RNA integrity was assessed by agarose gel electrophoresis, and DNAse treatment was applied due to the removal of contaminating genomic DNA from the RNA samples. Ten micrograms of RNA were diluted with 1 U of DNase I (M0303, New England Biolabs Inc, Ipswich, UK) and 10 μL of 10× DNAse buffer to a final volume of 100 µL upon the inclusion of DEPC water, for 15–20 min at 37 °C. Before the DNAse inactivation at 75 °C for 10 min, EDTA should be added to a final concentration of 5 mM to protect RNA from being degraded during enzyme inactivation. From each sample, 500 ng of total RNA was reverse-transcribed to cDNA using the Prime Script RT Reagent Kit (Perfect Real Time, Takara Bio Inc., Shiga-Ken, Japan) according to the manufacturer’s recommendations. All cDNAs were stored afterwards at −20 °C.

The following *Gallus gallus* genes were examined: aryl hydrocarbon receptor (*AhR*), AhR Nuclear Translocator (*ARNT*), cytochrome P450 (*CYP*) enzymes (*CYP1A1*, *CYP1A2*, *CYP1B1*), *NAD(P)H* quinone dehydrogenase 1 (*NQO1*), nuclear factor erythroid 2-like 2 (*Nrf2*), kelch-like ECH-associated protein 1 (*Keap1*), catalase (*CAT*), superoxide dismutase 1 (*SOD1*), glutathione peroxidase 2, 7 (*GPX2*, *GPX7*), heme oxygenase 1 (*HMOX1*), glutathione S-transferase alpha 2 (*GSTA2*), glutathione-disulfide reductase (*GSR*), peroxiredoxin-1 (*PRDX1*), glyceraldehyde 3-phosphate dehydrogenase (*GAPDH*), and actin beta (*ACTB*). Suitable primers were designed using the GenBank sequences deposited in the National Center for Biotechnology Information and US National Library of Medicine (*NCBI*) shown in Table 1.

Real-time PCR was performed in 96-well microplates with a SaCycler-96 Real-Time PCR System (Sacace Biotechnologies s.r.l., Como, Italy) and FastGene IC Green 2× qPCR universal mix (Nippon Genetics, Tokyo, Japan). Each reaction contained 12.5 ng RNA equivalents as well as 200 to 250 nmol/L of forward and reverse primers for each gene. The reactions were incubated at 95 °C for 3 min, followed by 40 cycles of 95 °C for 5 s, 60 or 62 °C (depending on the target gene) for 20 s, and 72 °C for 33 s. This was followed by a melt curve analysis to determine the reaction specificity. Each sample was measured in duplicate. Relative expression ratios of target genes were calculated according to an adapted multi-reference gene normalization procedure, using *GAPDH* and *ACTB* as reference genes.

### 2.4. Ethical Approval and Animal Welfare

The experimental protocol regarding the care and use of animals was approved by the Research Bioethics Committee of the Agricultural University of Athens, Greece (approval 33/240720). 

### 2.5. Statistical Analysis

Experimental data on layer performance were measured on a cage basis, whereas duodenal gene expressions were based on individual layers. All data were initially checked for normality and subsequently analyzed with the general linear model (GLM)–ANOVA procedure using the SPSS for Windows statistical package program, version 27 (SPSS Inc., Chicago, IL, USA). Statistically significant effects were further analyzed, and means were compared using Tukey’s honest significant difference (HSD) multiple comparison procedure. Statistical significance was determined at *p* ≤ 0.05. Linear (lin) and quadratic (quad) response patterns to dietary PP inclusion level were studied using polynomial contrasts.

## 3. Results

### 3.1. Growth Performance Responses

For the period of 1–8 experimental weeks (i.e., 21–28 weeks of layer age), laying rate, egg mass, feed intake, and feed conversion ratio did not differ between the treatments. The egg laying rate displayed a quadratic pattern of increase, with the P1000 treatment being the highest for this period (Table 2). For the period of 1–12 weeks, the laying rate (*p* < 0.001) in the P750 and P1000 treatments were significantly higher than in the CON. The egg mass (*p* = 0.008) was also significantly increased in the P1000 treatment compared with the CON. Moreover, the dietary PP supplementation resulted in a linear (*p* = 0.028) and quadratic (*p* < 0.01) increase in the egg laying rate and a quadratic (*p* = 0.005) increase in egg mass in the period of 1–12 weeks (Table 3).

### 3.2. Assessment of Gene Expressions in the Duodenum

#### 3.2.1. AHR Pathway

In the 8th week of the experiment (Table 3), the relative expression levels of *CYP1A1* (*p* < 0.001), *CYP1A2* (*p* < 0.001), and *CYP1B1* (*p* < 0.001) were significantly lower with PP administration. On the other hand, PP inclusion significantly (*p* < 0.001) upregulated relative expression levels of *GST* compared with the control. The gene expression of *ARNT* and *NQO1* was not significantly affected (*p* > 0.05) by PP inclusion. Moreover, the expression levels of *CYP1A1* (*P_lin_* < 0.001), *CYP1A2* (*P_lin_* < 0.001), and *CYP1B1* (*P_lin_* < 0.001) decreased linearly, whereas the expression of *GSTA2* increased linearly (*P_lin_* < 0.001). Polynomial contrast analysis showed that the relative expression of *CYP1A1*, *CYP1A2*, and *CYP1B1* displayed quadratic patterns of decrease with increasing PP inclusion level, whereas the relative expression of *AhR1* was increased (Table 4). 

In addition, in the 12th week the relative expression levels of *AhR* (*p* = 0.002), *ARNT* (*p* = 0.004), *CYP1A1* (*p* < 0.001), *CYP1A2* (*p* = 0.001), and *CYP1B1* (*p* = 0.012) were significantly downregulated compared with the control, whereas the *GST* (*p* = 0.002) and *NQO1* (*p* = 0.039) relative expression levels were significantly higher with the addition of PP. A linear pattern of decrease was displayed in the expression of *ARNT*, *CYP1A1*, *CYP1A2*, and *CYP1B1* with increasing PP level, while the expression level of *GSTA2* was increased (*P_lin_* = 0.016). The expression of *ARNT* (*P_quad_* = 0.005), *CYP1A1* (*P_quad_* < 0.001), *CYP1A2* (*P_quad_* = 0.011), and *CYP1B1* (*P_quad_* < 0.001) showed a quadratic pattern of decrease with increasing PP level, while the expression of *NQO1* (*P_quad_* = 0.004) increased (Table 5).

#### 3.2.2. NRf2/ARE Pathway

In the 8th week of the experiment, significant changes between the experimental treatments were observed in the expression of *NRf2* (*p* = 0.007), *SOD* (*p* = 0.049), *GPX7* (*p* = 0.005), *GSR* (*p* = 0.012), *PRDX1* (*p* = 0.008), *HMOX1* (*p* < 0.001), and *HSP90* (*p* = 0.011). However, there were no significant differences in the expression of *KEAP1* and *HSP70* between the treatments. The expression levels of *NRf2* (*P_lin_* = 0.022), *GPX2* (*P_lin_* = 0.021), and *GPX7* (*P_lin_* = 0.014) displayed linear patterns of increase whereas expression levels of *PRDX1* (*P_lin_* = 0.008) and SOD (*P_lin_* = 0.008) displayed linear patterns of decrease with increasing PP level. An increasing PP inclusion level resulted in patterns of increase in a quadratic manner for *CAT* (*P_quad_* = 0.009), *GPX7* (*P_quad_* = 0.002), and *HMOX1* (*P_quad_* < 0.001) (Table 6). 

Regarding the 12th week of the experiment, PP inclusion significantly upregulated the relative expression levels of *NRf2* (*p* = 0.004), *CAT* (*p* < 0.001), *SOD* (*p* = 0.021), *GPX2* (*p* = 0.039), *GSR* (*p* < 0.001), *PRDX1* (*p* = 0.008), and *HMOX1* (*p* = 0.05). Nevertheless, the expression level of *KEAP1* (*p* < 0.001) was significantly decreased. Moreover, the expression levels of *GPX7*, *HSP70*, and *HSP90* were not significantly affected (*p* > 0.05) by PP inclusion. Furthermore, an increasing PP inclusion level resulted in patterns of both increase and decrease in a linear and quadratic manner. More specifically, the *NRf2* (*P_lin_* = 0.019, *P_quad_* = 0.002), *CAT* (*P_quad_* < 0.001), *SOD* (*P_lin_* = 0.025, *P_quad_* = 0.032), *GSR* (*P_lin_* < 0.001), *PRDX1* (*P_lin_* = 0.001), and *HMOX1* (*P_quad_* = 0.013) expressions were higher compared with the control, whereas *KEAP1* (*P_lin_* < 0.001) expressions were lower (Table 7).

## 4. Discussion

This study aimed to extract new knowledge on the mechanisms of phytogenic function in the duodenum of laying hens under normal physiological and non-challenge experimental conditions. Thus, two critical pathways were monitored at the molecular level, namely, the AhR and Nrf2 pathways. These two pathways are relevant to detoxification and antioxidant capacity, respectively. In the gastrointestinal tract, it has been reported that AhR–Nrf2 interaction promotes detoxification by synergistically activating Phase I and II xenobiotic-metabolizing enzymes (XMEs) [5].

In particular, the inducible cellular cytoprotection is mediated by two signaling pathways, the AhR and Nrf2 pathways. The AhR pathway is responsible for the detoxification of xenobiotic compounds such as dioxins, mycotoxins and phytochemicals. The transcription factors *AhRs* exist as a multiprotein complex in the cytoplasm and bind xenobiotic AhR ligands entering the cell and subsequently translocating to the nucleus and heterodimerize with *AHR* nuclear translocator—*ARNT* [13]. Then, *AhR*/*ARNT* recognizes the xenobiotic-responsive elements (XREs) region of target genes known as Phase I xenobiotic-metabolizing enzymes (XMEs) and regulates their expression and downstream xenobiotic detoxification [14]. Specifically, the *AhR*-*ARNT* complex binds to XRE and regulates the expression of xenobiotic-metabolizing enzymes (XME) such as quinone oxidoreductase 1 (NQO1), glutathione transferase A2 (GSTA2), and cytochrome P450 (CYP) enzymes (CYP1A1, CYP1A2, CYP1B1). The quinone oxidoreductase 1 (NQO1) and glutathione transferase A2 (GSTA2) enzymes are both linked in the AhR and Nrf2 pathways, displaying detoxifying and antioxidant properties [15].

The Nrf2 pathway is responsible for the regulation of the antioxidant response and inflammation [8]. More specifically, *Nrf2* is in the cytoplasm and is bound with its inhibitor Kelch-like ECH-associated protein-1 (*Keap1*). Upon activation by ROS, *Nrf2* separates from *Keap1*, translocates to the nucleus, dimerizes with the small musculoaponeurotic fibrosarcoma protein (*sMAF*), and afterwards binds at the antioxidant response element (ARE) DNA regions of its target genes. This binding results in the transcription of Phase II antioxidant and cytoprotective genes, such as *CAT*, *SOD*, *GPX2*, *GPX7*, *HMOX1*, *GSTA2*, *GSR*, and *PRDX1* [9].

In our study, the trial started when the layers were in the beginning of the peak phase and lasted for 12 weeks. All data on the performance parameters were monitored closely on a weekly basis, and when the performance response started to differ, the layers’ intestines were sampled in week 8 and week 12 of the experiment. 

The zootechnical performance results of the present study showed that PP inclusion in the diets of laying hens increased the overall laying rate and egg mass, a finding that is in line with other studies using phytogenics [16,17]. It is well known that dietary phytogenics have beneficial effects on physiology, the metabolism of egg production, egg quality, and the general health status of birds [18]. In addition, beneficial effects of dietary phytogenics on gut health, the digestion of nutrients, and intestinal integrity have been reported [19,20,21]. These beneficial effects could be directly associated with improvements in productive performance. Moreover, the increase in the productive performance of hens could be attributed to phytogenics’ content of bioactive components, which have been shown to possess antimicrobial and antioxidant activities [22]. 

Furthermore, PP inclusion strongly modulated the expression of genes in the duodenum for both pathways. The study findings highlight that PP inclusion downregulated the *AhR* genes, whereas it upregulated the *Nrf2* genes. In particular, for the genes studied, PP inclusion decreased the expression of most detoxification genes (*ARNT*, *CYP1A1*, *CYP1A2*, *CYP1B1*) and increased the expression of most cytoprotective antioxidant and anti-inflammatory genes (*Nrf2*, *CAT*, *SOD*, *GPX2*, *GSR*, *PRDX1*, *HMOX1*) both in the 8th and 12th week of the experiment. 

According to several studies, when stress factors such as heat stress [23,24,25], mycotoxins [24,26], and pathogens [27,28] occur, the relevant expression of the AhR pathway genes increases while the relevant expression of *Nrf2*-related genes decreases. These are linked to a reduction in the productivity and intestinal health of the chickens [29]. Interestingly, the results of our study under non-challenge conditions demonstrated that the physiological host response to PP involved the beneficial modulation of the AhR/Nrf2-pathway-related genes supporting the improvements in the productivity of the layers. These findings could be an additional asset for laying hen performance and intestinal health under stressful conditions.

The aforementioned changes showed mainly a quadratic pattern of change, with increasing PP inclusion level, indicating the PP inclusion range level with optimal effects. Several studies have reported that the bioactive compounds of phytogenics demonstrate dose–response activities and are considered to be hormetic compounds, i.e., they induce biologically opposite effects at different doses [30,31]. Therefore, our findings supported and verified the demand of optimizing PP inclusion levels with respect to the targeted biological responses.

## 5. Conclusions

In conclusion, this study using a nutrigenomics approach has provided further mechanistic support for the PP benefits shown for layer performance. Phytogenic modulation of the layer *AhR*/*Nrf2* intestinal response has now been documented. This knowledge could provide a new basis for diet formulation strategies using phytogenics in commercial conditions with multiple stressor challenges that holds much promise and remains to be confirmed. 

## Figures and Tables

**Table 1 animals-13-00294-t001:** Oligonucleotide primers used for gene expression of selected targets by quantitative real-time PCR.

Target ^1^	Primer Sequence (5′-3′) ^2^	Annealing Temperature (°C)	PCR Product Size (bp)	GenBank (NCBI Reference Sequence)
*GAPDH*	F: ACTTTGGCATTGTGGAGGGT R: GGACGCTGGGATGATGTTCT	59.5	131	NM_204305.1
*ACTB*	F: CACAGATCATGTTTGAGACCTT R: CATCACAATACCAGTGGTACG	60	101	NM_205518.1
AhR pathway				
*AhR1*	F: TTTAGTGTGGCAGGTGGATT R: CCTTGTGCCAATGATGCTATTTG	60	200	NM_204118.2
*ARNT*	F: GAGACCAAGGCCCCAACTAC R: TCGGGTGCCTCTTTCTTTCC	62	140	NM_204200.1
*CYP1A1*	F: GTGATGGAGGTGACCATCGG R: ACATTCGTAGCTGAACGCCA	62	165	NM_205147.1
*CYP1A2*	F: CTGACCGTACACCACGCTT R: CTCGCCTGCACCATCACTTC	62	75	NM_205146.2
*CYP1B1*	F: CAGTGACTCCGCATCCCAAA R: CCATACGCTTACGGCAGGTT	62	132	XM_015283751.2
*GSTA2*	F: GCCTGACTTCAGTCCTTGGT R: CCACCGAATTGACTCCATCT	60	138	NM_001001776.1
*NQO1*	F: GAGCGAAGTTCAGCCCAGT R: ATGGCGTGGTTGAAAGAGGT	60.5	150	NM_001277619.1
Nrf2 pathway				
*Nrf2*	F: AGACGCTTTCTTCAGGGGTAG R: AAAAACTTCACGCCTTGCCC	60	285	NM_205117.1
*Keap1*	F: GGTTACGATGGGACGGATCA R: CACGTAGATCTTGCCCTGGT	62	135	XM_025145847.1
*CAT*	F: ACCAAGTACTGCAAGGCGAA R: TGAGGGTTCCTCTTCTGGCT	60	245	NM_001031215
*SOD1*	F: AGGGGGTCATCCACTTCC R: CCCATTTGTGTTGTCTCCAA	60	122	NM_205064.1
*GPX2*	F: GAGCCCAACTTCACCCTGTT R: CTTCAGGTAGGCGAAGACGG	62	75	NM_001277854.1
*GPX7*	F: GGCTCGGTGTCGTTAGTTGT R: GCCCAAACTGATTGCATGGG	60	139	NM_001163245.1
*GSR*	F: GTGGATCCCCACAACCATGT R: CAGACATCACCGATGGCGTA	62	80	XM_015276627.1
*HMOX1*	F: ACACCCGCTATTTGGGAGAC R: GAACTTGGTGGCGTTGGAGA	62	134	NM_205344.1
*PRDX1*	F: CTGCTGGAGTGCGGATTGT R: GCTGTGGCAGTAAAATCAGGG	61	105	NM_001271932.1
Heat Shock Proteins				
*HSP70*	F: ATGCTAATGGTATCCTGAACG R: TCCTCTGCTTTGTATTTCTCTG	60	145	NM_001006685.1
*HSP90*	F: CACGATCGCACTCTGACCAT R: CTGTCACCTTCTCCGCAACA	60	196	NM_001109785.1

^1^ *GAPDH* = glyceraldehyde 3-phosphate dehydrogenase; *ACTB* = actin beta; *AhR1* = aryl hydrocarbon receptor 1; *ARNT* = aryl hydrocarbon receptor nuclear translocator; *CYP1A1* = cytochrome P450 1A1; *CYP1A2* = cytochrome P450 1A2; *CYP1B1* = cytochrome P450 1B1; *GSTA2* = glutathione S-transferase alpha 2; *NQO1* = NAD(P)H quinone dehydrogenase 1; *Nrf2* = nuclear factor; erythroid 2-like 2; *Keap1* = kelch-like ECH-associated protein 1; *CAT* = catalase; *SOD1* = superoxide dismutase 1; *GPX2, 7* = glutathione peroxidase 2, 7; *GSR* = glutathione-disulfide reductase; *HMOX1* = heme oxygenase 1; *PRDX1* = peroxiredoxin-1; *HSP70* = heat shock 70 kDa protein; *HSP90* = heat shock protein 90 alpha family class A member. ^2^ F—Forward, R—Reverse.

**Table 2 animals-13-00294-t002:** Overall performance parameters of laying hens during weeks 1 to 8 of the experiment.

Data	Laying %	Egg Mass (g/hen/day)	Feed Intake (g/hen/day)	FCR
Treatments ^1^				
CON	95.40	54.68	106.1	1.94
P500	96.25	55.90	109.0	1.95
P750	97.03	55.29	107.7	1.95
P1000	97.34	56.22	108.7	1.94
P1500	95.46	54.61	107.7	1.97
Statistics ^2^				
SEM ^3^	0.790	0.762	1.50	0.027
*P_anova_*	0.062	0.162	0.369	0.492
*P_linear_*	0.499	0.919	0.402	0.433
*P_quadratic_*	0.008	0.051	0.182	0.507

^1^ PP supplementation (CON = 0 mg/kg, P500 = 500 mg/kg, P750 = 750 mg/kg, P1000 = 1000 mg/kg, and P1500 = 1500 mg/kg of diet). Data represent treatment means from n = 7 replicates per treatment (CON, P500, P750, P1000, and P1500). ^2^ Means with different superscripts (a, b) within the same column differ significantly (*p* ≤ 0.05). ^3^ Standard error of means.

**Table 3 animals-13-00294-t003:** Overall performance parameters of laying hens for weeks 1 to 12 of the experiment.

Data	Laying %	Egg Mass (g/hen/day)	Feed Intake (g/hen/day)	FCR
Treatments ^1^				
CON	95.44 ^a^	55.98 ^a^	107.9	1.94
P500	96.81 ^abc^	57.54 ^ab^	110.5	1.93
P750	97.70 ^bc^	57.05 ^ab^	109.6	1.93
P1000	98.27 ^c^	58.22 ^b^	110.3	1.91
P1500	96.16 ^ab^	56.63 ^ab^	110.0	1.96
Statistics ^2^				
SEM ^3^	0.557	0.585	1.28	0.027
*P_anova_*	<0.001	0.008	0.321	0.066
*P_linear_*	0.028	0.142	0.222	0.871
*P_quadratic_*	<0.001	0.005	0.211	0.254

^1^ PP supplementation (CON = 0 mg/kg, P500 = 500 mg/kg, P750 = 750 mg/kg, P1000 = 1000 mg/kg, and P1500 = 1500 mg/kg of diet). Data represent treatment means from n = 7 replicates per treatment (CON, P500, P750, P1000, and P1500). ^2^ Means with different superscripts (a, b, c) within the same column differ significantly (*p* ≤ 0.05). ^3^ Standard error of means.

**Table 4 animals-13-00294-t004:** Relative expression of AhR pathway genes in the layers’ duodenum at the 8th week of the experiment.

Genes	Treatments ^1^					Statistics ^2^			
Duodenum	CON	P500	P750	P1000	P1500	SEM ^3^	*P_anova_*	*P_linear_*	*P_quadratic_*
AhR pathway									
*AhR1*	0.94	1.17	1.33	1.28	0.77	0.292	0.282	0.732	0.036
*ARNT*	1.76	1.25	0.90	1.65	1.87	0.526	0.346	0.601	0.077
*CYP1A1*	4.38 ^C^	2.33 ^B^	1.26 ^A^	0.42 ^A^	0.41 ^A^	0.372	<0.001	<0.001	<0.001
*CYP1A2* ^4^	3.51 ^C^	1.90 ^BC^	0.40 ^A^	1.03 ^AB^	0.47 ^A^	0.396	<0.001	<0.001	0.011
*CYP1B1* ^4^	3.42 ^C^	2.39 ^BC^	0.62 ^AB^	0.56 ^AB^	0.44 ^A^	0.467	<0.001	<0.001	0.037
*GSTA2*	0.46 ^A^	1.06 ^B^	1.25 ^B^	1.52 ^B^	2.10 ^B^	0.269	<0.001	<0.001	0.978
*NQO1*	1.25	1.35	1.31	0.93	0.91	0.270	0.317	0.079	0.419

^1^ PP supplementation (CON = 0 mg/kg, P500 = 500 mg/kg, P750 = 750 mg/kg, P1000 = 1000 mg/kg, and P1500 = 1500 mg/kg of diet). Data represent treatment means from n = 7 replicates per treatment (CON, P500, P750, P1000, and P1500). ^2^ Means with different superscripts (A, B, C) within the same row differ significantly (*p* ≤ 0.05). ^3^ Standard error of means. ^4^ Data for CYP1A2 and CYP1B1 were transformed to ln.

**Table 5 animals-13-00294-t005:** Relative expression of the AhR pathway genes in the layers’ duodenum at the 12th week of the experiment.

Genes	Treatments ^1^					Statistics ^2^			
Duodenum	CON	P500	P750	P1000	P1500	SEM ^3^	*P_anova_*	*P_linear_*	*P_quadratic_*
AhR pathway									
*AhR1*	1.16 ^B^	1.01 ^AB^	0.79 ^A^	1.09 ^AB^	1.03 ^AB^	0.137	0.002	0.519	0.058
*ARNT* ^4^	2.56 ^B^	0.99 ^A^	0.99 ^A^	0.60 ^A^	0.92 ^A^	0.422	0.004	<0.001	0.005
*CYP1A1*	2.88 ^D^	1.55 ^C^	1.03 ^BC^	0.65 ^AB^	0.41 ^A^	0.196	<0.001	<0.001	<0.001
*CYP1A2*	1.78 ^B^	0.73 ^A^	0.98 ^A^	0.95 ^A^	0.87 ^A^	0.231	0.001	0.004	0.011
*CYP1B1* ^4^	2.44 ^B^	0.68 ^A^	0.79 ^A^	0.91 ^A^	0.91 ^A^	0.372	0.012	0.004	<0.001
*GSTA2*	0.67 ^A^	1.53 ^AB^	0.96 ^AB^	1.39 ^AB^	1.68 ^B^	0.327	0.002	0.016	0.861
*NQO1*	0.94 ^A^	1.18 ^AB^	1.24 ^AB^	1.35 ^B^	0.90 ^A^	0.163	0.039	0.809	0.004

^1^ PP supplementation (CON = 0 mg/kg, P500 = 500 mg/kg, P750 = 750 mg/kg, P1000 = 1000 mg/kg, and P1500 = 1500 mg/kg of diet). Data represent treatment means from n = 7 replicates per treatment. ^2^ Means with different superscripts (A, B, C, D) within the same row differ significantly (*p* ≤ 0.05). ^3^ Standard error of means. ^4^ Data for *ARNT* and *CY1B1* to ln.

**Table 6 animals-13-00294-t006:** Relative expression of Nrf2 pathway genes and heat shock response-related genes in the layers’ duodenum at 8th week of the experiment.

Genes	Treatments ^1^					Statistics ^2^			
Duodenum	CON	P500	P750	P1000	P1500	SEM ^3^	*P_anova_*	*P_linear_*	*P_quadratic_*
Nrf2 pathway									
*Nrf2*	0.60 ^A^	1.23 ^AB^	2.20 ^B^	1.56 ^AB^	1.90 ^AB^	0.540	0.007	0.022	0.134
*KEAP1*	0.93	0.94	1.25	1.03	0.93	0.229	0.590	0.859	0.226
*CAT*	1.19	1.92	2.12	1.78	0.57	0.597	0.087	0.309	0.009
*SOD*	1.48 ^B^	1.36 ^AB^	1.14 ^AB^	1.27 ^AB^	0.73 ^A^	0.250	0.049	0.008	0.466
*GPX2*	0.80	0.90	1.01	1.93	1.50	0.446	0.257	0.021	0.845
*GPX7*	0.65 ^A^	1.06 ^AB^	1.59 ^B^	1.40 ^B^	1.13 ^AB^	0.235	0.005	0.019	0.002
*GSR*	1.17 ^A^	1.06 ^A^	1.53 ^AB^	1.91 ^B^	1.20 ^AB^	0.319	0.012	0.209	0.135
*PRDX1*	1.42 ^AΒ^	1.57 ^Β^	1.23 ^AΒ^	1.27 ^AΒ^	0.63 ^A^	0.296	0.008	0.008	0.135
*HMOX1* ^4^	0.73 ^A^	2.63 ^B^	1.15 ^A^	1.16 ^A^	0.77 ^A^	0.288	<0.001	0.110	<0.001
Heat Shock Response									
*HSP70*	0.91	0.80	1.48	1.44	1.12	0.303	0.115	0.131	0.160
*HSP90*	1.10 ^AB^	1.10 ^AB^	1.31 ^AB^	1.41 ^B^	0.85 ^A^	0.251	0.011	0.743	0.070

^1^ PP supplementation (CON = 0 mg/kg, P500 = 500 mg/kg, P750 = 750 mg/kg, P1000 = 1000 mg/kg, and P1500 = 1500 mg/kg of diet). Data represent treatment means from n = 7 replicates per treatment. ^2^ Means with different superscripts (A, B) within the same row differ significantly (*p* ≤ 0.05). ^3^ Standard error of means. ^4^ Data for HMOX1 were transformed to ln.

**Table 7 animals-13-00294-t007:** Relative expression of Nrf2 pathway genes and heat shock response-related genes in the layers’ duodenum at the 12th week of the experiment.

Genes	Treatments ^1^					Statistics ^2^			
Duodenum	CON	P500	P750	P1000	P1500	SEM ^3^	*P_anova_*	*P_linear_*	*P_quadratic_*
Nrf2 pathway									
*Nrf2*	0.63 ^A^	1.18 ^AB^	1.60 ^B^	1.26 ^AB^	1.21 ^AB^	0.220	0.004	0.019	0.002
*KEAP1*	1.74 ^B^	0.98 ^A^	1.12 ^AB^	0.66 ^A^	0.78 ^A^	0.219	<0.001	<0.001	0.052
*CAT*	0.60 ^A^	1.27 ^BC^	1.22 ^BC^	1.42 ^C^	0.85 ^AB^	0.154	<0.001	0.071	<0.001
*SOD*	0.73 ^A^	0.97 ^AB^	1.55 ^AB^	1.88 ^B^	1.20 ^AB^	0.351	0.021	0.025	0.032
*GPX2*	0.96 ^A^	0.95 ^A^	1.31 ^AB^	1.39 ^B^	1.10 ^AB^	0.167	0.039	0.064	0.066
*GPX7*	0.99	1.12	0.99	1.21	1.11	0.147	0.531	0.341	0.765
*GSR*	0.54 ^A^	1.12 ^AB^	1.14 ^AB^	1.59 ^B^	1.46 ^B^	0.213	<0.001	<0.001	0.092
*PRDX1*	0.76 ^A^	1.19 ^AB^	1.29 ^AB^	1.55 ^B^	1.38 ^B^	0.205	0.008	0.001	0.066
*HMOX1* ^4^	0.70 ^A^	1.41 ^B^	1.35 ^B^	1.31 ^B^	1.14 ^AB^	0.246	0.050	0.167	0.013
Heat Shock Response									
*HSP70*	1.15	0.96	1.24	1.50	0.97	0.216	0.103	0.727	0.223
*HSP90*	1.12	1.01	0.92	1.03	1.16	0.180	0.708	0.830	0.167

^1^ PP supplementation (CON = 0 mg/kg, P500 = 500 mg/kg, P750 = 750 mg/kg, P1000 = 1000 mg/kg, and P1500 = 1500 mg/kg of diet). Data represent treatment means from n = 7 replicates per treatment. ^2^ Means with different superscripts (A, B, C) within the same column differ significantly (*p* ≤ 0.05). ^3^ Standard error of means. ^4^ Data for HMOX1 were transformed to ln.

## Data Availability

The data analyzed in this study are available from the corresponding author on reasonable request.

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
