# Peer review of "Phytogenic Effects on Layer Production Performance and Cytoprotective Response in the Duodenum"

_animals, 2023, doi:10.3390/ani13020294_

Round 1

Reviewer 1 Report

General Comments;

-        Nice paper with a much-needed research approach for the underlying issues in layer welfare with antibiotics becoming obsolete within the poultry industry. I just have a few questions/comments that need to be addressed before in can approve for publication. Thank you!

-        Line 12; change “on production performance and critical detoxification and antioxidant molecular responses” to “on production performance, critical detoxification and antioxidant molecular responses”

-        Line 17; add a comma after approach

-        Line 60; add the after at intestinal level

-        Line 68; change “on production performance and on chicken critical detoxification (AhR pathway) and antioxidant…” to “on production performance, on chicken critical detoxification (AhR pathway) and antioxidant…”

-        Line 264; reference?

-        Line 268’ change week 8th to week 8

Thank you!

Reviewer 2 Report

Due to many uncertainties regarding the methodology and the research material used in the calculations (see comments in the text), I ended the evaluation of the manuscript in the Results section at this stage. I cannot verify the experiment's correctness, including from the statistical point of view!

Round 2

Reviewer 2 Report

All comments were introduced in the text.
